# Controlling a Van Hove singularity and Fermi surface topology at a complex oxide heterostructure interface

Ryo Mori [1,2], Patrick B. Marshall[3], Kaveh Ahadi[3], Jonathan D. Denlinger[4], Susanne Stemmer[3] & Alessandra Lanzara[1,5]*

The emergence of saddle-point Van Hove singularities (VHSs) in the density of states, accompanied by a change in Fermi surface topology, Lifshitz transition, constitutes an ideal ground for the emergence of different electronic phenomena, such as superconductivity, pseudo-gap, magnetism, and density waves. However, in most materials the Fermi level, $E_F$, is too far from the VHS where the change of electronic topology takes place, making it difficult to reach with standard chemical doping or gating techniques. Here, we demonstrate that this scenario can be realized at the interface between a Mott insulator and a band insulator as a result of quantum confinement and correlation enhancement, and easily tuned by fine control of layer thickness and orbital occupancy. These results provide a tunable pathway for Fermi surface topology and VHS engineering of electronic phases.

[1] Materials Sciences Division, Lawrence Berkeley National Laboratory, Berkeley, CA 94720, USA. [2] Applied Science & Technology, University of California, Berkeley, CA 94720, USA. [3] Materials Department, University of California, Santa Barbara, CA 93106-5050, USA. [4] Advanced Light Source, Lawrence Berkeley National Laboratory, Berkeley, CA 94720, USA. [5] Department of Physics, University of California, Berkeley, CA 94720, USA. *email: alanzara@lbl.gov

Quantum interactions at the interfaces between strongly correlated materials trigger new emergent phenomena that do not exist in each single material alone[1–10]. Their properties can be controlled by varying the parent constituents and/or their stacking geometry[5,9–14]. One notable example is the emergence of electron liquid behavior, created by novel charge mismatch in the quantum well of nonpolar band insulator $SrTiO_3$ embedded in polar Mott insulator, such as $SmTiO_3$, showing hallmarks of strongly correlated phases, including non-Fermi liquid behavior[15,16], transport lifetime separation[16], pseudo-gap[17], spin density waves[15], and antiferromagnetism[18]. These behaviors are tuned by the thickness of the quantum wells in $SrTiO_3$, defined by the number of SrO layers ($t_{QW} = x$-SrO), embedded in $SmTiO_3$. Transport measurements have shown two different regimes as a function of thickness. Specifically, the temperature dependence of resistivity shows a sharp change from a $T^2$ behavior to a $T^n$ (with $1 < n < 2$) behavior as the number of layers decreases, with a sharp crossover at the critical thickness $t_{QW} \sim 5$-SrO where a divergence of resistivity and other electronic properties, such as the Hall coefficient, is observed[15–18]. While the origin of these exotic properties is believed to be related to quantum confinement, involving atomic orbital and coupling/correlation of the interfacial electron liquids, no study of the electronic structures at these buried interfaces exists.

Here, we use angle-resolved photoemission spectroscopy (ARPES) to access this information in $SrTiO_3$ band insulator embedded in Mott insulator $SmTiO_3$. As the critical thickness is approached[15,16], we reveal the formation of a Van Hove singularity (VHS), a change of orbital character and Fermi surface topology, reminiscent of a Lifshitz transition. These results allow us to identify the critical parameters for Fermi surface topology and VHS engineering of electronic phases at the interfaces.

## Results

**Fermi surface and orbital character.** Fig. 1a illustrates the four types of structures, $t_{QW} = \infty$, 6, 5, and 3-SrO, investigated in this work. The role of $SmTiO_3$ is to create a potential barrier and charge source to confine the electronic states within $SrTiO_3$ and prevent their diffusion in the $SmTiO_3$ layers[19]. Each structure belongs to three distinct transport regimes: $T^2$ regime ($n \sim 2$) for the $\infty$- and 6-SrO layer, $1 < n < 2$ regime for 3-SrO layers and the critical crossover regime for 5-SrO layers[15–17]. Figure 1b–i shows the evolution of the Fermi surfaces as a function of thickness for two different photon polarizations. Given the fact that both $SrTiO_3$ and $SmTiO_3$ are insulating and the experimental evidences shown and discussed in the "Method" section and the Supplementary Notes 1–3, these Fermi surfaces are representative of the buried interface conducting states. For $p$–polarized light (Fig. 1 b–e), the main feature is an elliptical electron pocket at the Brillouin zone (BZ) center (most visible in the 2nd BZ, see green arrows). Following previous works[20–27], we associate these pockets with the Ti $d_{yz/xz}$ orbitals. The intensity variation across the different BZ is due to photoemission matrix element effects: $p$–polarized light is mostly sensitive to orbitals with even symmetry with respect to the scattering plane. As the number of SrO layers decrease, we observe an increase in the size of the electron pocket. This suggests an increase of electron occupancy in the $d_{yz/xz}$ orbitals. When the light polarization is changed from $p$ to $s$, additional circular features appear both at the BZ center and corner (see red arrows in Fig. 1f–i). These states have odd parities relative to the horizontal plane. The pockets at the BZ center resemble similar features observed in the electron gas of $SrTiO_3$[22–24] and can be associated with the subband states of Ti $d_{xy}$ orbitals[20–27].

The additional circular pockets at the BZ corner are present in neither bulk $SrTiO_3$ nor $SmTiO_3$[19,23]. Because $SmTiO_3$ has the orthorhombic distorted perovskite structure with $TiO_6$ octahedral rotation, these new structures suggest the presence of structural reconstruction at the interface, resulting in Fermi surface folding ($\sqrt{2} \times \sqrt{2} \times 45°$)[28–30]. Note that the finite intensity around X point (best observed in Fig. 1e) is relatively less clear than the folded $d_{xy}$ state at M, but may suggest the possibility of a $2 \times 2$ in-plane reconstruction instead as also reported for $SrTiO_3/LaAlO_3$ heterostructure[27]. Similar reconstructions are observed in all the studied thicknesses, suggesting a negligible effect of the distortion between different samples. In contrast to the elliptical features, the size of the circular pockets decreases from $\infty$-SrO to $t_{QW} \leq 5$-SrO, suggesting a decrease of occupancy of the $d_{xy}$ orbitals. This can be seen in more details in Fig. 1j–m, where the two splitting states of the $d_{xy}$ pocket in $\infty$-SrO are clearly resolved. This splitting is considered to be the subband states or a Rashba like splitting of the $d_{xy}$ pocket, a manifestation of spin–orbit interaction, similarly observed in electron gas on the surface of $SrTiO_3$[22–24,31].

From the Fermi surface area we find that the charge density increases from $(1.9 \pm 0.1) \times 10^{14}$ cm$^{-2}$ ($\infty$-SrO), $(2.7 \pm 0.7) \times 10^{14}$ cm$^{-2}$ (6-SrO), $(4.7 \pm 0.4) \times 10^{14}$ cm$^{-2}$ (5-SrO), to $(5.4 \pm 0.2) \times 10^{14}$ cm$^{-2}$ for 3-SrO. As the quantum well thickness decreases, the total mobile charge is confined in a narrower region, making the carrier density in each $TiO_2$ layer higher, consistent with the trend in our results. We note that the estimated carrier density from our ARPES data may be slightly increased, due to the presence of additional carriers originating from oxygen vacancies induced by annealing and synchrotron radiation (See Supplementary Note 1 for more details) as reported in previous works[22,23]. However, our results are in consistent well with those expected within the transport measurements, given the ambiguity in the interpretation of the Hall coefficient[15,16,32,33].

**Electronic dispersion and mass renormalization.** Figure 2 shows the electronic dispersion, energy versus momentum of each orbital, for the different quantum well structures. In general, the data for the $\infty$-SrO (Fig. 2a, e, i) resemble the surface and interface states of $SrTiO_3$[21–24,34–39]. The top and middle panels (Fig. 2a–h) show the dispersion for the $d_{yz/xz}$ electron pockets, while the bottom panels show the dispersion for the $d_{xy}$ states. The bandwidth of the $d_{yz/xz}$ electron pockets (Fig. 2a–d and e–h) increases from $\sim$38 meV for the $\infty$-SrO, $\sim$85 meV for 6-SrO to $\sim$130 meV for 5-SrO, and decreases to a slightly shallower level $\sim$110 meV for the 3-SrO system. In contrast, the bandwidth of the $d_{xy}$ electron pockets (Fig. 2i–l) decreases in accordance with the decrease of the Fermi surface area ($\sim$260 meV ($\sim$150 meV) for the outer (inner) pocket in the $\infty$-SrO, $\sim$190 meV for the 6-SrO, $\sim$165 meV for the 5-SrO, and $\sim$195 meV for the 3-SrO).

In order to quantify the electron effective mass (electron correlation) for each orbital, the band renormalization factor, $Z$, is determined from the renormalized dispersion $E$:

$$E(k_{||}) = Z E_{DFT}(k_{||}) + \varepsilon_0 = Z E_{Bare}(k_{||}), \qquad (1)$$

where $E_{DFT}$ is the band dispersion of the bulk $SrTiO_3$ Ti $t_{2g}$ orbitals obtained by first principle calculations (DFT), and $E_{Bare}$ is the bare band (black solid line in Fig. 2) based on the experimental Fermi momentum $k_F$ and the bandwidth $\varepsilon_0$ (See "Method" section for more detail). As shown in Fig. 2, the renormalized band dispersions $E$ (green dashed line) sufficiently match the ARPES spectra. In general, the larger discrepancy between the renormalized band dispersions and the bare band

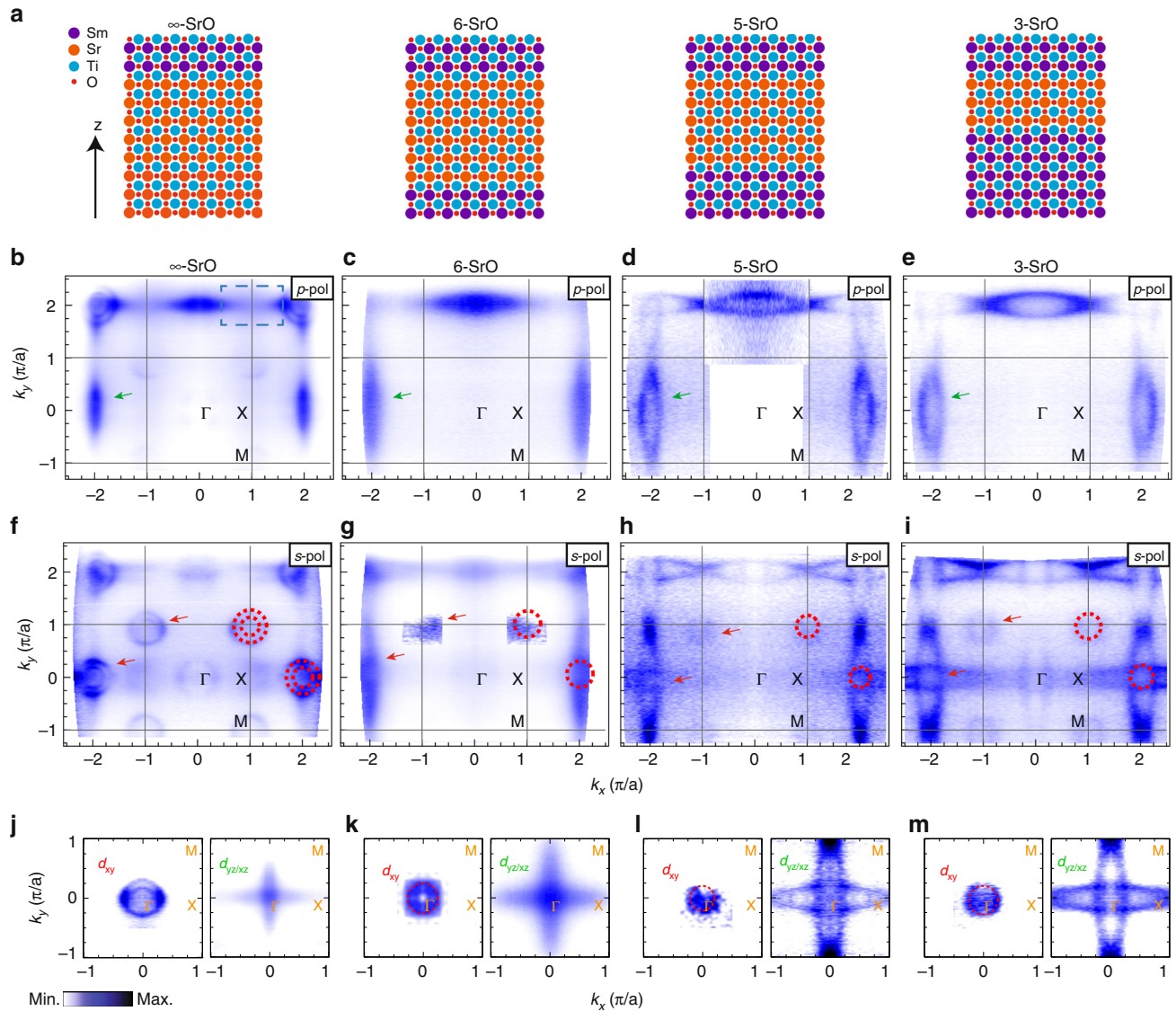

**Fig. 1** Overview of the heterostructures and evolution of the interface Fermi surfaces. **a** The schematics of heterostructures studied in this work. Under the top capping layers of SmTiO$_3$ (two quasi-unit cells), $x$ layers of SrTiO$_3$ are embedded. The thickness of SrTiO$_3$ $t_{QW}$ is defined by the number of SrO layers ($x$-SrO). From left to right, $t_{QW} = \infty$, 6, 5, and 3-SrO structures are shown, respectively. Fermi surface maps at $E_F$ for different quantum well states taken over multiple BZ: $t_{QW} = \infty$ (**b, f**), 6 (**c, g**), 5 (**d, h**), and 3 (**e, i**)). Spectra are obtained by integrating intensities in the energy window of $\pm 10$ meV at $E_F$. Data are taken with two polarization ($p$–polarization in **b**–**e**, and $s$–polarization **f**–**i**,). The black solid line represents the in-plane BZ of SrTiO$_3$ with high symmetry points marked. The green arrows (red circulars and red arrows) are guides to the eye for the Fermi surface of $d_{yz/xz}$ ($d_{xy}$) orbital. Zoom in of Fermi surface structures for each Ti $d$ orbital, $d_{xy}$ (left) and $d_{yz/xz}$ (right) for $t_{QW} = \infty$ (**j**), 6 (**k**), 5 (**l**), and 3 (**m**), obtained by symmetrization and/or summarization of equivalent BZ. The dashed red circular contours are guides to the eye as the Fermi surfaces of $d_{xy}$ orbital. For a direct comparison, all Fermi surfaces shown here are centered at $(k_x, k_y) = (0, 0)$, and shown in a range corresponding to the BZ of SrTiO$_3$.

(black solid line) indicates greater mass enhancement factors (1/Z). The effective mass of $d_{yz/xz}$ orbital bands clearly shows significant enhancement for the 5- and 3-SrO systems, while effective masses in $\infty$- and 6-SrO are described as the bare band mass ($Z \simeq 1$). Such enhancement of the mass renormalization and hence electron correlation can be explained by the increase of electron doping in each TiO$_2$ layers. Indeed, as the distance between interfaces decreases, more electrons are confined in the narrower and flatter quantum potential, resulting in a significant increase of electrostatic doping level.

**Lifshitz transition and emergence of Van Hove singularity.** In Fig. 3, we study the details of the electron pocket and Fermi surface topology near the BZ boundary (the X point, see blue

dashed square in Fig. 1b) for the different layers. For the $\infty$-SrO (Fig. 3a), the $d_{yz/xz}$ state forms a closed Fermi surface, forming an isolated small electron pocket centered around the Γ point as also seen in Fig. 1b, and hence absence of electronic states in the proximity of the X point (see also the dispersion curves in Fig. 3k, l). Once the thickness of SrTiO$_3$ changes to 6-SrO, the $d_{yz/xz}$ state evolves to have a larger Fermi surface and a deeper bandwidth as discussed in Figs. 1 and 2, and the topology of the Fermi surface is the same as the $\infty$-SrO (a closed electron pocket centered around the X point). On the contrary, as the number of layers decreases $t_{QW} \leq 5$SrO, the size of the electron pocket at the Γ point expands eventually covering the entire BZ, till the X point, see Fig. 3c, d. This gives rise to a hole-like structure along Γ-X direction (light-blue dashed line), and electron-like structure along the other

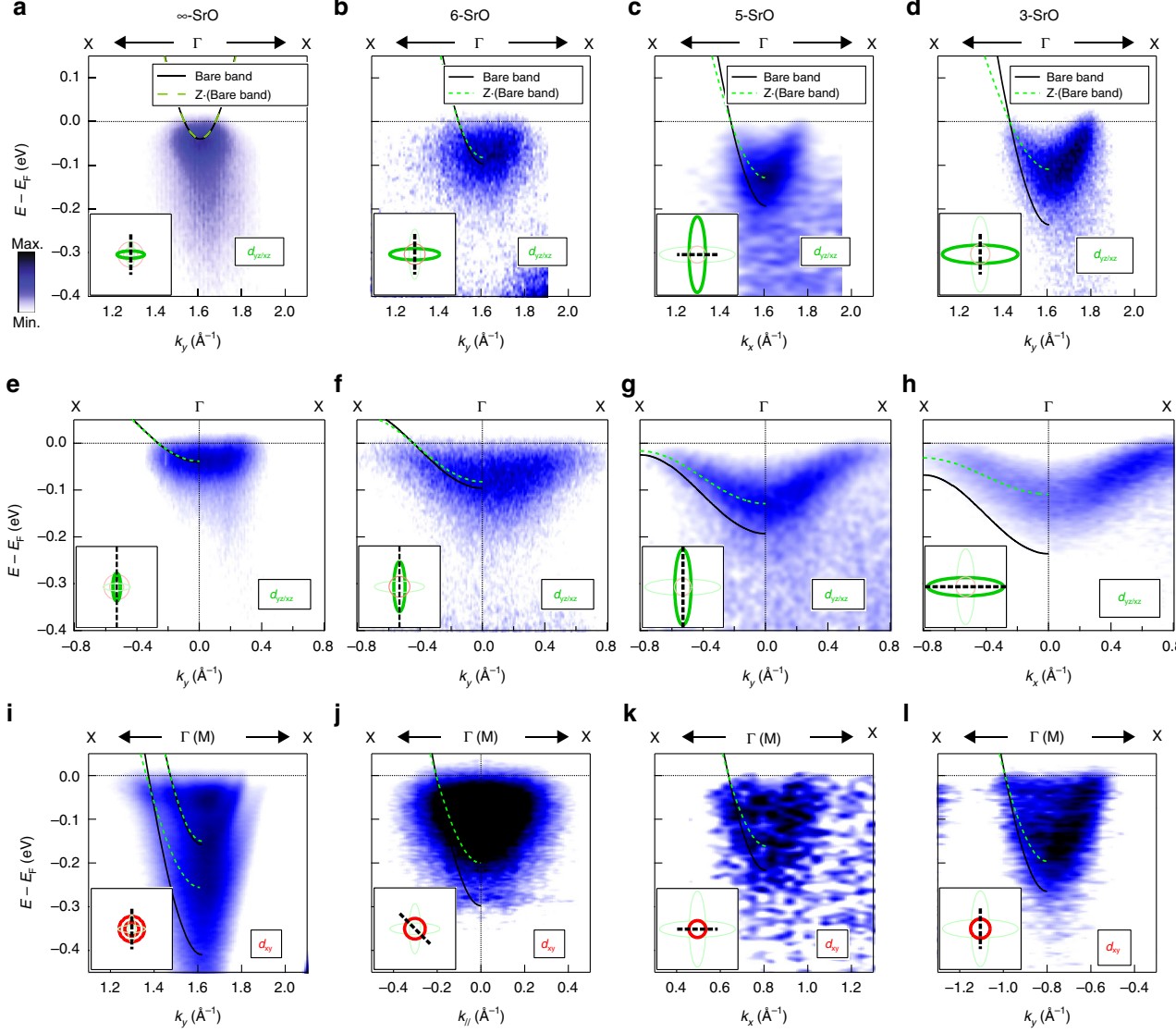

**Fig. 2** Evolution of electronic structures at the interfaces for different quantum well structures. ARPES spectra for light band $d_{yz/xz}$ orbital ($t_{QW} = \infty$- (**a**), 6- (**b**), 5- (**c**), and 3- (**d**) SrO). ARPES spectra for heavy band $d_{yz/xz}$ ($t_{QW} = \infty$- (**e**), 6- (**f**), 5- (**g**), and 3- (**h**) SrO). ARPES spectra of energy versus momentum cuts for $d_{xy}$ orbital as a function of number of layer ($t_{QW} = \infty$- (**i**), 6- (**j**), 5- (**k**), and 3- (**l**) SrO). Only $\infty$-SrO shows clear evidence of two parabolic sub-bands. The insets in each figure show the locations of the cuts. The black solid lines and green dashed lines are the bare bands based on DFT of bulk $SrTiO_3$ and the renormalized experimental band dispersions, respectively.

direction Γ-M (red dashed line). The detailed Fermi surface structure around the X point is shown in Fig. 3h, i. The peak positions in the momentum-distribution curves (MDC) in Fig. 3h, i and the MDCs along M-X-M shown in Fig. 3j delineate a clear change in Fermi surface topology. The dashed lines #1–6 in Fig. 3f–i indicate the directions of energy versus momentum cuts along M-X-M (#1, 3, 5, and 7) and Γ-X-Γ (#2, 4, 6, and 8) presented in Fig. 3k–r. The peak positions of MDC (red dots in Fig. 3n, o, q) and energy-distribution curve (EDC) (light-blue dots in Fig. 3n–r) reveal electron- and hole-like dispersion at the X point. Moreover, as shown in Fig. 3o–r clearly, the local minimum and local maximum are observed at the same momentum location, the X point, displaying the emergence of a saddle-point VHS, where the curvatures of bands have opposite signs along two orthogonal directions (schematically illustrated in Fig. 3e). Figure 3s shows the EDCs divided by the Fermi–Dirac function at the X point for each structure indicating the evolution of energy level of VHS.

## Discussion

In summary, by studying the evolution of the thickness dependence of the electronic states in $SrTiO_3$ confined by Mott insulator $SmTiO_3$, we report a change of Fermi surface topology from a closed electron pocket centered around the Γ point for $t_{QW} > 5$-SrO, to a large-opened electron pocket extending all the way to the BZ boundary for $t_{QW} \leq 5$-SrO as schematically summarized in Fig. 4a. As the distance between the two interfaces decreases, the interface starts to couple each other drastically when the distance of the interfaces is smaller than $t_{QW} \sim 6$-SrO (see Supplementary Note 5), resulting in the change of orbital occupancy order as shown in Fig. 4c. In this regime ($t_{QW} \leq 6$-SrO), as the quantum well thickness decreases, the interface coupling increases, resulting in higher doping and hence deeper bandwidth. $t_{QW} \leq 5$-SrO layer appears to be a critical thickness beyond which the interlayer coupling reach a critical value leading to increase of electron–electron interaction and therefore a strong renormalization, as shown in Fig. 2 and summarized in

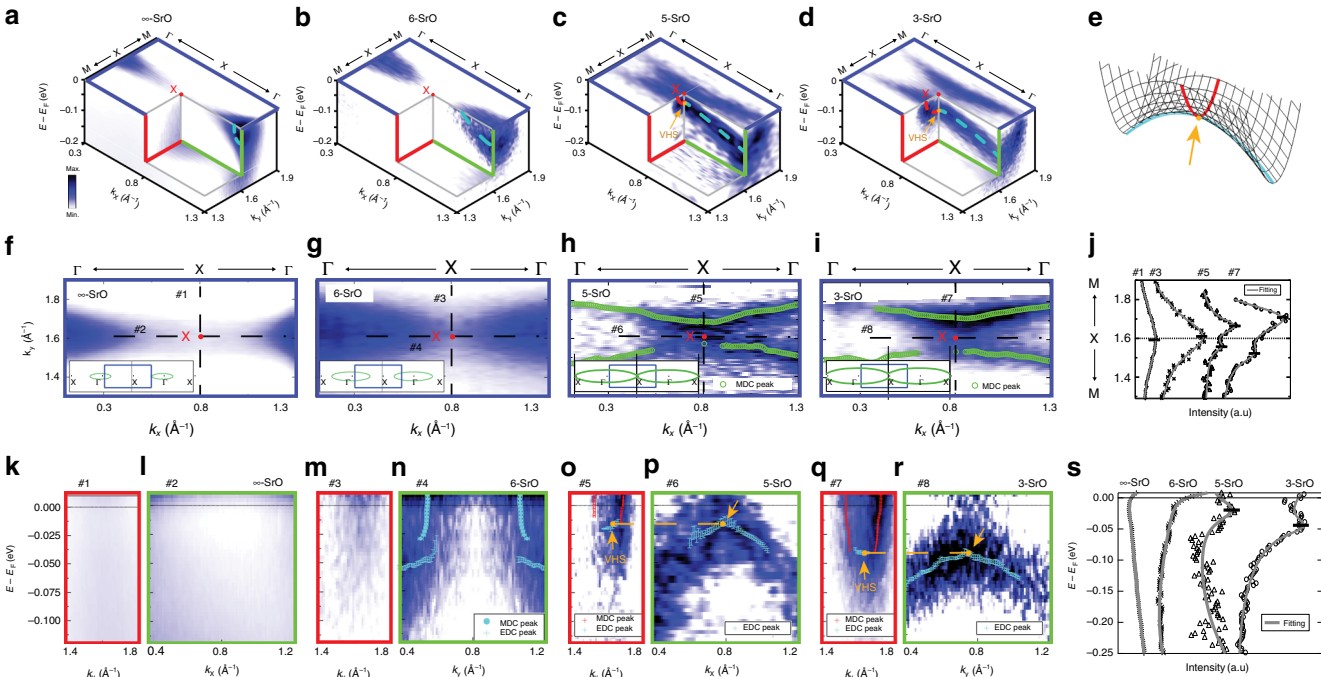

**Fig. 3** Topology change in Fermi surface and emergence of Van Hove singularity. 3D ARPES spectra around the X point for $\infty$-SrO (**a**), 6-SrO (**b**), 5-SrO (**c**), and 3-SrO (**d**), including the symmetrized Fermi surfaces relative to the $d_{yz}$ states (blue squares) and the energy and momentum cuts along M-X-M (red squares) and $\Gamma$-X-$\Gamma$ (green squares). The red and light-blue dashed lines are guides to eyes, displaying the dispersion along M-X-M and $\Gamma$-X-$\Gamma$, respectively. The yellow dot and arrow in **c**, **d** indicate the location of the Van Hove singularity (VHS) at X points. **e** A schematic of a general saddle-point singularity. The yellow dot and arrow indicate the saddle-point where the dispersion has the local minimum in one direction (red) and local maximum in another direction (light-blue). The Fermi surface around X point for $\infty$-SrO (**f**), 6-SrO (**g**), 5-SrO (**h**), and 3-SrO (**i**), respectively. The blue squares in the insets show the locations of the Fermi surfaces. Peak positions, obtained by fitting with Lorentzian curves along $k_y$-direction, are shown in **h** and **i** with green dots superposed. The error bar of peak positions is smaller than the size of dots. **j** MDCs at $E_F$ along M-X-M (cuts #1, #3, #5, and #7 in **f-i**,) for $\infty$-SrO (#1), 6-SrO (#3), 5-SrO (#5), and 3-SrO (#7) with fittings (gray lines) from left to right. For a direct comparison, all MDCs here are normalized based on the intensities at $k_y$ = 1.3. **k-r** The energy and momentum cuts for cuts #1–8 as shown in **f-i**. All spectra shown here are divided by the energy resolution convolved Fermi-Dirac function (FD) at the measured temperature. The peak positions in the MDC and EDC are denoted by red (light-blue filled circles in **n**) and light-blue cross dots, respectively, in **n-r**. The yellow dashed line and arrows in **o-r** are guides to the eyes to indicate the location of the VHS at X points. The error bars in **n-r** represent the uncertainties of peak positions from Lorentzian fits. **s** FD divided EDCs around X points for $\infty$-SrO, 6-SrO, 5-SrO, and 3-SrO with fittings (gray lines) from left to right. All EDCs are normalized for comparison.

Fig. 4d. The change of Fermi surface topology, accompanied by the emergence of a VHS is the typical features of a Lifshitz transition[40] and is the result of an increase of electron correlation[41–43]. Further theoretical studies, such as DFT + DMFT, are needed to investigate the direct relation between the correlation enhancement and Lifshitz transition in the oxide interface system. In addition, such transition may also explain the reported sudden change in resistivity temperature dependence for $t_{QW} \leq$ 5-SrO, shown in Fig. 4b and coincides with the divergence of Hall coefficient[16]. The change in Fermi surface topology is also supported by the change of the orbital occupancy for $t_{QW} \leq$ 5-SrO, as shown in Fig. 4c. The biggest change is observed for the $d_{yz/xz}$ orbitals, mainly associated with the electron pockets centered at $\Gamma$. This orbital configuration change is in qualitative agreement with recent DFT calculations on the Fermi surface reconstruction by strain and confinement in similar systems[44]. We note also that the relation between effective masses and occupancy for $d_{xy}$ orbital bands are in good agreement with previous studies, suggesting a similar potential well of the surface states on SrTiO₃ (interface of SrTiO₃/vacuum)[22,23,39].

The emergence of a VHS near the Fermi level $E_F$ on the electric properties has been discussed for past decades, especially because of the induced electronic instability leading unconventional phenomena, such as superconductivity, pseudo-gap, magnetism, and density waves[45–50]. Manipulation of the position of VHS near $E_F$ is shown

to be the key to give rise to a new phase of matter. However, changing the energy level of VHS near low-energy level has been impossible for most materials. Here, as shown in Fig. 3o–s, the position of VHS in the SmTiO₃/SrTiO₃ can be tuned in low-energy region by changing the number of layers, $t_{QW}$, as a result of quantum confinement and couplings discussed above. These results suggest that the quantum confinement at the oxide interface provides a robust possibility for engineering electronic phases. Finally, it is noteworthy to point out the obtained locations with respect to $E_F$ of VHS in 5-SrO and 3-SrO system ($\sim$15 and $\sim$40 meV, respectively) are in scale with the recent observation of pseudo-gap, $2\Delta \sim$ 65 meV in 2-SrO[17]. Further studies are needed to investigate in details the relation between the two.

In summary, we discovered a change of Fermi surface topology, typical of a Lifshitz transition occurs at the interface between a Mott insulator and a band gap insulator by changing quantum confinement of the 2D electron liquid. Moreover, we find that such transition is accompanied by the appearance of a VHS and an enhancement of electron correlation. These results provide a new tunable pathway to engineer Fermi surface topology and VHSs in materials.

## Methods

**Samples.** All samples were grown on (001) (LaAlO₃)$_{0.3}$(Sr₂AlTaO₆)$_{0.7}$ (LSAT) substrates by hybrid molecular beam epitaxy. The quantum well thickness and the

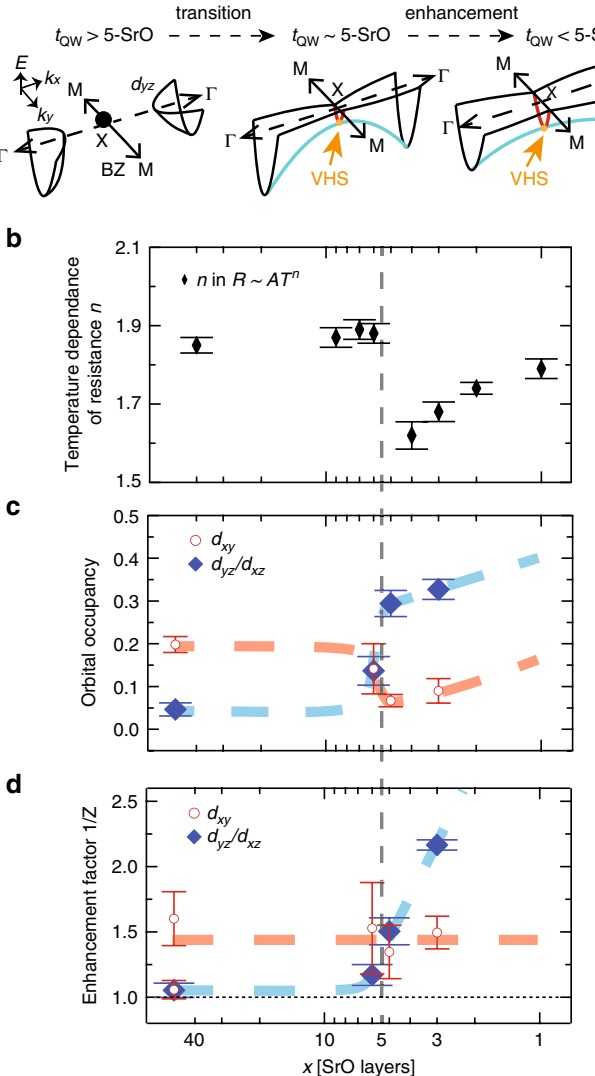

The samples were annealed at 700–800 °C for 30–60 min in ultra high vacuum (UHV), transferred to the ARPES chamber in-situ, and measured at 20–25 K in UHV better than $3 \times 10^{-11}$ Torr using photon energy of 80–120 eV and a Scienta R4000 analyzer. All data shown in the main text were taken with photon energy of 92 eV. The energy resolution was 20–30 meV and the angular resolution was better than 0.2° for all measurements. It should be noted that the following arguments corroborate that our ARPES results are mainly produced by the buried interfaces. First, because the top layers of $SmTiO_3$ of all samples have the same thicknesses and structure, they should not account for a significant difference in the electronic structures between the samples. Second, although the details of surface conditions between the measurements might be different based on the low-energy electron diffraction (LEED), the main results near the $E_F$ shown in this work were reproduced from the same samples but different LEED quality (Supplementary Note 2). This suggests that the data in this work are less sensitive to the surface, but more to the interface which is protected by the capping layer. In addition, the observation of $Ti^{+4}$ core level spectra ensures that the detected photoelectrons come from the buried $SrTiO_3$ layers as the Ti layers in $SmTiO_3$ consist of only $Ti^{+3}$ (Supplementary Note 1). Thus, we believe that the difference in the electronic structure observed in this work stem mainly from the interface.

**Theory**. Theoretical calculations were performed within generalized gradient approximation (GGA) for bulk $SrTiO_3$ as implemented in the QUANTUM ESPRESSO Package[53,54]. Correlation effects are taken into account through an on-site Hubbard $U$ correction in form of GGA + $U$, with the local Coulomb repulsion $U$ for Ti $3d$ electrons being 4.5 eV. Ultra-soft pseudopotentials with a plane wave energy cutoff of 30 Ry were used, and Monkhorst k-mesh was chosen as $32 \times 32 \times 32$. The crystal structure considered in the calculation is simple cubic with 3.905 Å of the lattice constant. The calculated electronic structures for Ti $dt_{2g}$ orbitals ($E_{DFT}(k_{||})$) have been used to evaluate the experimental electronic structures. The bare band structure $E_{Bare}(k_{||})$ is obtained by using the following equation:

$$E_{Bare}(k_{||}) = E_{DFT}(k_{||}) - E_{DFT}(k_F), \qquad (2)$$

where $k_F$ is the experimental Fermi momentum, obtained by fitting with Lorentzian function to MDC at $E_F$. The band renormalization factor $Z$ is obtained from: $Z = \varepsilon_0/E_{DFT}(k_F)$ where $\varepsilon_0$ is the experimental bandwidth. For the renormalization factor for heavy $d_{yz/xz}$ bands (especially for 5- and 3-SrO), we use the same value obtained by light $d_{yz/xz}$ bands.

## Data availability

The data that support the finding of this study are available from the corresponding author upon request.

**Fig. 4** Fermi surface topology and interlayer coupling **a** The schematics of Lifshitz transition and the effect of interlayer coupling as a function of $t_{QW}$. The red and light-blue lines highlight the electron- and hole-like dispersion, respectively. After Lifshitz transition, the electron correlations in the quantum well are enhanced by the effect of interface–interface coupling, making the Van Hove singularity (VHS) deeper and the bandwidth around the Γ point shallower. **b** The temperature dependence of the resistance ($R \sim AT^n$) from the transport measurement in refs. [16], showing the phase transition. **c** Evolution of orbital occupancy for orbital ($d_{xy}$ (red circle) and $d_{yz/xz}$ (blue diamond) orbitals as a function of $t_{QW}$. The total occupancy of the $d_{yz/xz}$ orbitals is twice of the value shown here, because of the degeneracy. **d** Electron mass enhancement factor $1/Z$ for each orbital, $d_{xy}$ (red circle) and light $d_{yz/xz}$ (blue diamond) as a function of $t_{QW}$. The red and blue dashed lines in **c** and **d** are guides to the eye for other thickness structures. The vertical black dashed line in **b**–**d** separates the two distinct phases of the system[15,16]. The error bars in **c**, **d** are estimated by absolute maximum variations of peak positions obtained by Lorentzian fittings for EDC and MDC.

absence of strain relaxation/dislocations were confirmed by using transmission electron microscopy . The pseudocubic lattice parameter of $SmTiO_3$ is about 3.9 Å. The detail of the growth can be found in refs. [51,52]. To measure the interface directly, only two quasi-unit cells of top layer $SmTiO_3$, which acts as the capping layer to both protect and create the interface states at once, are used.

**Photoemission measurement**. ARPES measurements were performed at the Beamline 4.0.3 (MERLIN) of the Advanced Light Source in Berkeley, California.

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

## Acknowledgements

We thank P. Krüger and D.-H. Lee for useful discussions. This work was primarily funded by the U.S. Department of Energy (DOE), Office of Science, Office of Basic Energy Sciences, Materials Sciences and Engineering Division under Contract No. DE-AC02-05-CH11231 (Quantum materials KC2202) and used resources of the Advanced Light Source, a DOE Office of Science User Facility under contract no. DE-AC02-05CH11231. A.L. and R.M. acknowledge partial support for this research from the Gordon and Betty Moore Foundation's EPiQS Initiative through grant GBMF4859. Sample growth was supported by National Science Foundation under Grant No. 1740213 (P.B.M., K.A, S.S.). R.M. also acknowledges support from the Funai Foundation for Information Technology.

## Author contributions

A.L. and S.S. initiated and directed this research project. The samples were grown and characterized by P.B.M. and K.A. ARPES measurements were carried out by R.M. with the assistance of J.D.D. R.M. analyzed the data, did the calculation, and wrote the text with feedback from all authors. All authors contributed to the scientific planning and discussions.

## Competing interests

The authors declare no competing interests.
