## [Peer Review File · Nature Communications]

Reviewers' comments:

Reviewer #1 (Remarks to the Author):

In this work, Mori and coworkers carried out ARPES experiments on heterostructures consisting of SmTiO₃ and SrTiO₃ (SrO) with SrO's thickness varied as 3, 5 and infinity. It is claimed that such heterostructures go through the topological transition in the momentum space, i.e. Lifshitz transition, when the SrO thickness is varied from 5 to infinity. I do not have any concern on the quality of the experimental data presented. However, I feel that the presented data is insufficient to support their main claim. After completing the data set, this paper could be published in some journal. Unfortunately, however, I do not think this paper belongs to a high impact journal such as Nature Communications.

Certainly, the band topology is one of the central issues in the current condensed matter physics in connection with a variety of topological insulators. Therefore, if one could control the Lifshitz transition or the Fermi surface topology in topological materials, that would be extremely interesting. However, I do not see such novelty in this work. I do not think similar methods could be used to control the band topology in topological materials.

A couple of questions and suggestions.

It is not clear at what SrO thickness the Lifshitz transition really takes place. The current paper only shows that 3-SrO and 5-SrO have open Fermi surfaces for dyz/dxz bands, and infinite(!)-SrO has closed Fermi surfaces for dyz/dxz bands. The authors argue that the critical SrO thickness is 5 based on the temperature dependence of the resistance from Ref. 22. I do not think this claim is well supported by the ARPES data in the current manuscript. Quite surprisingly, Ref. 22 from the same group argued that the QCP in this system does not appear to be a Lifshitz transition (because the residual resistance does not show a discontinuity across the transition). Therefore, unless the authors provide additional ARPES data for at least 6-SrO, showing closed Fermi surfaces for dyz/dxz bands, this paper is incomplete. However, even if the authors could provide such data, I do not think it would add the novelty to warrant the publication on Nature Communications.

I do not understand the statement on the relation between the mass enhancement and the occupancy of dxy bands. How is it in good agreement with Refs, 26, 43 and 45?? Ref. 43 uses only DFT and TB, therefore there is no mass enhancement.

If electron correlations are enhanced by VHS, influencing the resistance, there would be a signature in ARPES spectra, not just mass enhancement but the broadening of the quasiparticle peak. In particular, if the non-Fermi liquid state is realized in 3-SrO and 5-SrO, quasiparticles are not well defined at least at some momenta. Where are such effects?

Since the authors have access to a DFT code, I would suggest carrying out more realistic calculations considering interfaces between SrTiO₃ and SmTiO₃ because the comparison between bulk dispersions and ARPES dispersions of oxide interfaces to discuss mass enhancement does not seem to be right; dyz/dxz and dxy bands are hybridized and therefore, if their relative level is changed, their dispersions are changed in a complex manner.

Reviewer #2 (Remarks to the Author):

The manuscript presents ARPES results of electronic band structure in the interface between Mott insulator (SmTiO₃) and a band insulator (SrTiO₃). The position of Van Hove singularity (VHS) is found to shift in energy depending on the thickness of SrTiO₃ layers sandwiched by SmTiO₃ layers, which consequently varies the topology of Fermi surface (FS). In addition, the authors

reveal the variation of band width likely due to the correlation effect for different SrTiO₃-layers, and claim that this is the origin for the discontinuous variation of transport behavior such as resistivity with the thickness of SrTiO₃ layers.

Bellow are my concerns which prevent the publication of this paper.

1) It is not clear why the authors could conclude that their ARPES data are taken from the interface; it is possible that the signature actually comes from the bulk. The authors should measure the photon energy dependence of FS, and confirm that the k_z dispersion of it is indeed absent, especially for the infinite layer system. This argument is critical for the main conclusion in this paper.

2) The most critical issue is that the results for 6-SrO (or 7-SrO) are missing in this paper, thus one cannot tell whether or not the jump of transport values seen across the border between 5-SrO and 6-SrO (Fig. 4b) really has a direct relation with the Lifshitz Transition of FS topology as claimed by the authors. I am insisting this since the role of VFs (and its energy position) on the physics of transition metal oxides is not so clear as in the introduction part, while there are many theories to suggest it. The best such example would be cuprates: the energy position of VFs is not the same as the end point of T_c dome. Moreover, cuprates do not show a discontinuous behavior in transports due to the Lifshitz Transition of FS topology in the overdoped regime; rather, a smooth variation with doping is observed, whereas Hall coefficient changes its sign for an obvious reason. To justify that the Lifshitz Transition of FS topology is certainly critical in the current system, the authors need to show that 6-SrO (or 7-SrO) has VFs above E_F differently from that for 5-SrO.

3) The data quality of Fig.2 is not so high, while the band width and related correlation effect are argued based on these. I suggest that the authors plot EDCs extracted from the ARPES images and demonstrate how the energy dispersions have been determined with these data.

4) The authors should more elaborately discuss what makes the energy shift of VFs with the thickness of SrTiO₃ layers, and why the electron correlations are related to it (or the topology of FS).

Minor issue: The green dotted FSs drawn in Fig. 1 is confusing, since it contradicts to the claim by the authors that 5-SrO and 3-SrO have open FSs (not closed ones) owing to VFs unoccupied.

In conclusion, the data set in this paper is incomplete not only to claim that the ARPES spectra excited only from the interface, but also to conclude that the energy location of VFs is critical for the discontinuous behavior of resistivity. Therefore, I cannot recommend this manuscript for publication in Nature communications.

Reply to the Report to Referee #1

*In this work, Mori and coworkers carried out ARPES experiments on heterostructures consisting of SmTiO_3 and SrTiO_3 (SrO) with SrO 's thickness varied as 3, 5 and infinity. It is claimed that such heterostructures go through the topological transition in the momentum space, i.e. Lifshitz transition, when the SrO thickness is varied from 5 to infinity. I do not have any concern on the quality of the experimental data presented. However, I feel that the presented data is insufficient to support their main claim. After completing the data set, this paper could be published in some journal. Unfortunately, however, I do not think this paper belongs to a high impact journal such as *Nature Communications*.*

Certainly, the band topology is one of the central issues in the current condensed matter physics in connection with a variety of topological insulators. Therefore, if one could control the Lifshitz transition or the Fermi surface topology in topological materials, that would be extremely interesting. However, I do not see such novelty in this work. I do not think similar methods could be used to control the band topology in topological materials.

First of all, we would like to clarify that the change in topology of Fermi surface shown in this work should not be considered in the same context as the topological insulators where the band inversions exist. The concept of Fermi surface topology is a general concept in condensed matter physics that reflects a change of the Fermi surface from hole-like to electron-like (or *vice versa*), often associated with the proximity to a Van Hove singularity (VHS), defined as the point where the curvature of the electronic bands as opposite sign in two orthogonal directions. Such change of topology can lead to a variety of new phenomena, and has been a central theme in condensed matter physics in the past few years.

Electronic correlations, including lattice, charge, and spin interactions, strongly depends on the topology of a given electronic structures. Graphene and cuprates are great examples where VHS and the topology of Fermi surface transition play an important role for the unique properties. These fields have been attracting great interests as witnessed by the high impact that these works have had in the literatures[1–6]. Moreover, the recent studies on moiré twisted bilayer graphene[7, 8], which are considered to be one of the most important studies nowadays, have suggested the important role of VHS[9, 10] in the unconventional behaviors. Our work shows that such conditions can be engineered in a controlled way at the interface between different oxides and could be responsible for the appearance of a variety of the rich and intriguing phenomena reported. Given these examples, we believe that our results will become a milestone in the years to come for both theory and experiment in the field of oxide science. Therefore, we believe that this work is worth to publish in *Nature Communications*.

A couple of questions and suggestions.

It is not clear at what SrO thickness the Lifshitz transition really takes place. The current paper only shows that 3SrO and 5SrO have open Fermi surfaces for dyz/dxz bands, and infinite(!) SrO has closed Fermi surfaces for dyz/dxz bands. The authors argue that the critical SrO thickness is 5 based on the temperature dependence of the resistance from Ref. 22. I do not think this claim is well supported by the ARPES data in the current manuscript. Quite surprisingly, Ref. 22 from the same group argued that the QCP in this system does not appear to be a Lifshitz transition (because the residual resistance does not show a discontinuity across the transition). Therefore, unless the authors provide additional ARPES data for at least 6SrO , showing closed Fermi surfaces for dyz/dxz bands, this paper is incomplete. However, even if the authors could provide such data, I do not think it would

add the novelty to warrant the publication on Nature Communications.

Thank you for the suggestion. We have now added the study of the 6-SrO layer sample to the manuscript. As it can be seen from the data, the Fermi surface for the 6 layer sample is closed around the Γ point as for the ∞ layer sample (see Fig. 1 and Fig. 3 of revised manuscript). The electronic structure is in agreement with a higher doping level, and shows a change of the occupancy in the vicinity of the thickness where the resistance shows a sudden change (\sim 5-SrO). These new results support the previous conclusion that a change in Fermi surface topology and emergence of VHS occur at the thickness where transport has identified a sudden change. Additionally, we show that such transition is accompanied by an enhancement of electron correlation and a drastic change of orbital occupancy.

I do not understand the statement on the relation between the mass enhancement and the occupancy of d_{xy} bands. How is it in good agreement with Refs, 26, 43 and 45?? Ref. 43 uses only DFT and TB, therefore there is no mass enhancement.

Thank you for pointing out this mistake about the reference. We have now changed the reference accordingly. We keep the Wang, Z. *et al* (Ref 45 in the previous manuscript. The reference number has changed to Ref 49 in the present form) and added the Meevasana, W. *et al* (Ref 31 in the present form) and the Santander-Syro, A.F. *et al* (Ref 32 in the present form) instead of the Chang, Y.J. *et al* (Ref 26 in the previous manuscript) and Zhong, Z *et al* (Ref 43 in the previous manuscript).

Our observation of a constant effective mass despite changes of Fermi momenta (Fermi surface area or occupancy) is in agreement with the previous reports[11–13] (Ref 49, 31, 32 in the present manuscript). In fact, as discussed in Ref[11] (Ref 49 in the main text), the effective mass of the two-dimensional electron gas varies with the Fermi wavevector k_F or orbital occupancy of the d_{xy} orbital and saturates for k_F above $\sim 0.143 \text{ \AA}^{-1}$. The Fermi momenta of the d_{xy} orbital states observed in this work are $\sim 0.25 \text{ \AA}^{-1}$ (∞ -SrO), $\sim 0.2 \text{ \AA}^{-1}$ (6-SrO), $\sim 0.17 \text{ \AA}^{-1}$ (5-SrO), and $\sim 0.19 \text{ \AA}^{-1}$ (3-SrO), respectively, well above $\sim 0.143 \text{ \AA}^{-1}$.

If electron correlations are enhanced by VHS, influencing the resistance, there would be a signature in ARPES spectra, not just mass enhancement but the broadening of the quasiparticle peak. In particular, if the nonFermi liquid state is realized in 3SrO and 5SrO, quasiparticles are not well defined at least at some momenta. Where are such effects?

Following the referee's request, in Fig. R1 we show the comparison between the MDCs at E_F of each orbital in the ∞ -SrO and 3-SrO samples. While the peak width of the outer (inner) MDCs for ∞ -SrO is $0.08 \pm 0.008 \text{ \AA}^{-1}$ ($0.09 \pm 0.005 \text{ \AA}^{-1}$), the width for 3-SrO is broader than ∞ -SrO, $0.17 \pm 0.01 \text{ \AA}^{-1}$. This change in the line width may suggest its transition from Fermi liquid to non-Fermi liquid. However, we would like to caution that there are several factors that could contribute to MDC broadening, such as surface defects and capping layer of SmTiO_3 . While there is no reason to believe that these are different among the studied samples, because of similar growth condition and procedure, we feel that such comparison should be taken with the benefit of doubt, and this is the reason why we have omitted from the manuscript and have instead focused on more reliable parameters such as mass enhancement. Indeed, most of the ARPES works in the literature study MDC width as a function of energy, measured on the same surface, rather than a comparison between

FIG. R1. MDCs at E_F for the d_{xy} states **a,b** MDC at E_F of the d_{xy} states for ∞ -SrO (**a**) and 3-SrO (**b**). Red curves are raw data and blue curves are fitted Lorentzian curves for each peak.

different surfaces.

Since the authors have access to a DFT code, I would suggest carrying out more realistic calculations considering interfaces between SrTiO₃ and SmTiO₃ because the comparison between bulk dispersions and ARPES dispersions of oxide interfaces to discuss mass enhancement does not seem to be right; dyz/dxz and dxy bands are hybridized and therefore, if their relative level is changed, their dispersions are changed in a complex manner.

Thank you for your suggestion. Indeed, there may be the hybridizations in a “complex” manner and we probably need deeper studies in both theoretical (such as DFT+DMFT[14]) and experimental works to explain the relation between our observation and other unusual phenomena, such as pseudo-gap. However, unfortunately, such “realistic” calculations on these complex systems are not only beyond our capabilities, but also beyond the main point and conclusion in this work. In this work, we basically compare the electron effective masses for each orbital between the different structures experimentally with reference to the bulk SrTiO₃. The obtained mass enhancement factors directly reflect the strength of the electron correlations of corresponding orbitals in each structure. Similar characterizations have already been done in the literature for different materials, such as SrVO₃[15, 16] and other SrTiO₃-based electron liquid[11]. Therefore, we believe that our characterization quantify the electron correlation in this complex system.

We thank again Referee #1 for the insightful feedbacks and suggestions. We hope that the Referee #1 will agree with the “novelty” of our work.

Reply to the Report to Referee #2

The manuscript presents ARPES results of electronic band structure in the interface between Mott insulator (SmTiO₃) and a band insulator (SrTiO₃). The position of Van Hove singularity (VHs) is found to shift in energy depending on the thickness of SrTiO₃ layers sandwiched by SmTiO₃ layers, which consequently varies the topology of Fermi surface (FS). In addition, the authors reveal the variation of band width likely due to the correlation effect for different SrTiO₃ layers, and claim that this is the origin for the discontinuous variation of transport behavior such as resistivity with the thickness of SrTiO₃ layers.

First of all, we thank the reviewer for the fruitful suggestions and comments. Below are our responses to all the comments:

Bellow are my concerns which prevent the publication of this paper.

1) It is not clear why the authors could conclude that their ARPES data are taken from the interface; it is possible that the signature actually comes from the bulk. The authors should measure the photon energy dependence of FS, and confirm that the k_z dispersion of it is indeed absent, especially for the infinite layer system. This argument is critical for the main conclusion in this paper.

We thank the referee for rising this point. In response to his/her comment, we have now added the photon energy dependence to the Supplementary information (Fig. S2) as well as a detailed description both in the method section and in the Supplementary Information. The following argument are in support of our conclusion that the states near E_F are coming from the interface.

1) Both SmTiO₃ and SrTiO₃ are insulators, with bandgap of ~ 2.2 eV and ~ 3.2 eV. Therefore, no states near the Fermi level are expected from each one of these bulk states. Therefore, the observed near E_F states can only be due to the SmTiO₃'s surface states or the buried interface. The photon energy dependence for the ∞ -SrO sample show non-dispersive d_{xy} states (see Fig. S3 in the Supplementary Information). Non-dispersive k_z feature is indicative of a two-dimensional surface state rather than a bulk states. The $d_{xz/yz}$ states show a dispersive feature along k_z even in the ∞ -SrO sample, due to the penetration of the wave function associated with these orbitals along the out-of-plane direction as discussed in the literature[11, 13, 17–20].

2) Surface states are very sensitive to surface conditions, this is obviously less the case for an interfacial state, protected by the capping layers. As surface quality decreases these states should become broader and eventually the dispersion hard to determine. Our experiments were conducted over multiple sample surfaces of different quality (see LEED patterns in Fig. S2 of the Supplementary Information). Despite the very different surface quality, the dispersions near E_F were reproduced in each of these samples, without no substantial differences observed. This further strengthens the idea that these states near E_F come from the interface. Moreover, the top layers of SmTiO₃ have the same thicknesses and structure for all the sample studied, and therefore cannot account for the significant differences observed in the electronic structure and the Fermi surface topology as a function of thickness. These observations suggest that the observed states result from the buried interface and are indicative of a doped SrTiO₃ state, also in agreement with literature reports[11–13, 21]. Moreover, the core level spectra (shown in Fig. S1 of the Supplementary Information) show that their main contribution comes from photoelectron in the Ti⁺⁴ core level. These core

levels are associated with SrTiO₃ and not SmTiO₃, as the TiO₂ layers in SmTiO₃ consists of only Ti⁺³ core level (see also discussion in the Supplementary Information and Ref[22]).

2) The most critical issue is that the results for 6SrO (or 7SrO) are missing in this paper, thus one cannot tell whether or not the jump of transport values seen across the border between 5SrO and 6SrO (Fig. 4b) really has a direct relation with the Lifshitz Transition of FS topology as claimed by the authors. I am insisting this since the role of VHS (and its energy position) on the physics of transition metal oxides is not so clear as in the introduction part, while there are many theories to suggest it. The best such example would be cuprates: the energy position of VHS is not the same as the end point of Tc dome. Moreover, cuprates do not show a discontinuous behavior in transports due to the Lifshitz Transition of FS topology in the overdoped regime; rather, a smooth variation with doping is observed, whereas Hall coefficient changes its sign for an obvious reason. To justify that the Lifshitz Transition of FS topology is certainly critical in the current system, the authors need to show that 6SrO (or 7SrO) has VHS above EF differently from that for 5SrO.

As requested by the referee, we have now measured a 6 layer sample (6-SrO) and have now added the data and discussion to the manuscript. We find that the overall topology of the $d_{yz/xz}$ electronic structure is similar to the ∞ -SrO with a closed Fermi surface centered at the Γ point and a VHS above E_F . The differences in the electronic structure (shown in Fig. 2 and 3 of the manuscript) are in agreement with a higher doping level for the 6-SrO sample and are in line with a sudden change of the occupancy in the vicinity of the thickness where the resistance shows a sudden change (\sim 5-SrO). These new data support our conclusion that the Lifshitz transition occurs near the border between 5 and 6 layers sample. We thank the referee for bringing this to our attention and have requested additionally data.

3) The data quality of Fig.2 is not so high, while the band width and related correlation effect are argued based on these. I suggest that the authors plot EDCs extracted from the ARPES images and demonstrate how the energy dispersions have been determined with these data.

Thank you for your suggestion. In addition to modifying Fig. 2 in the main text, we have added to the Supplementary Information (see Fig. S4) the EDCs extracted from our ARPES data and their fitting results for each orbital. In this Fig. S4, we also show the second derivatives of the ARPES intensities for each structure.

4) The authors should more elaborately discuss what makes the energy shift of VHS with the thickness of SrTiO3 layers, and why the electron correlations are related to it (or the topology of FS).

We thank the referee for this important comment. We have now added this discussion to the main text. The estimated quantum well potential width of each interface is \sim 13 Å (see Fig. S5 in the Supplementary Information), and hence the interface starts to couple when the distance of the interfaces is smaller than \sim 26 Å, i.e. $t_{QW} \sim$ 6-SrO (\sim 23.4 Å), resulting in a change of the orbital occupancy as shown in Fig. 4c of the main text. In this regime ($t_{QW} \leq$ 6-SrO), as the thickness of SrTiO₃ decreases, the interface coupling increases resulting in an increase of electrostatic doping at the interface. $t_{QW} \leq$ 5-SrO layer appears

to be a critical thickness beyond which the interlayer coupling reach a critical value leading to increase of electron-electron interaction and therefore a strong renormalization, as shown in Fig. 2 and summarized in Fig. 4d. The competition between these two effects (doping vs renormalization from electron correlation) makes the energy level of VHS is fine tunable in low-energy level.

Minor issue: The green dotted FSs drawn in Fig. 1 is confusing, since it contradicts to the claim by the authors that 5SrO and 3SrO have open FSs (not closed ones) owing to VHS unoccupied.

Thank you for pointing it out. We deleted the green dotted Fermi surface contours in Fig. 1.

In conclusion, the data set in this paper is incomplete not only to claim that the ARPES spectra excited only from the interface, but also to conclude that the energy location of VHS is critical for the discontinuous behavior of resistivity. Therefore, I cannot recommend this manuscript for publication in Nature communications

We thank the referee for the constructive comments and suggestions. We hope that with all the revision the paper is now suitable for publication in *Nature Communication*.

-
- [1] He, Y. et al. Fermi surface and pseudogap evolution in a cuprate superconductor. Science **344**, 608 (2014). URL <http://science.sciencemag.org/content/344/6184/608.abstract>.
 - [2] Fujita, K. et al. Simultaneous transitions in cuprate momentum-space topology and electronic symmetry breaking. Science **344**, 612 (2014). URL <http://science.sciencemag.org/content/344/6184/612.abstract>.
 - [3] Bragana, H., Sakai, S., Aguiar, M. C. O. & Civelli, M. Correlation-driven lifshitz transition at the emergence of the pseudogap phase in the two-dimensional hubbard model. Phys. Rev. Lett. **120**, 067002 (2018). URL <https://link.aps.org/doi/10.1103/PhysRevLett.120.067002>.
 - [4] Doiron-Leyraud, N. et al. Pseudogap phase of cuprate superconductors confined by fermi surface topology. Nature Communications **8**, 2044 (2017). URL <https://doi.org/10.1038/s41467-017-02122-x>.
 - [5] Kang, M. et al. Evolution of charge order topology across a magnetic phase transition in cuprate superconductors. Nature Physics (2019). URL <https://doi.org/10.1038/s41567-018-0401-8>.
 - [6] Li, G. et al. Observation of van hove singularities in twisted graphene layers. Nature Physics **6**, 109 (2009). URL <https://doi.org/10.1038/nphys1463>.
 - [7] Cao, Y. et al. Correlated insulator behaviour at half-filling in magic-angle graphene superlattices. Nature **556**, 80 (2018). URL <http://dx.doi.org/10.1038/nature26154>.
 - [8] Cao, Y. et al. Unconventional superconductivity in magic-angle graphene superlattices. Nature **556**, 43 (2018). URL <http://dx.doi.org/10.1038/nature26160>.
 - [9] Li, K.-Q. Y. L.-J. W. W.-X. Y. W. Y. X.-Q. Y. J.-K. L. H. J. H. H. L., Si-Yu; Liu. Evidence of electron-electron interactions around van hove singularities of a graphene moire superlattice. ArXiv e-prints (2017). 1609.04149.

- [10] Kerelsky, A. et al. Magic angle spectroscopy (2018). URL <https://ui.adsabs.harvard.edu/#abs/2018arXiv181208776K>.
- [11] Wang, Z. et al. Tailoring the nature and strength of electron-phonon interactions in the SrTiO₃(001) 2D electron liquid. Nat Mater **15**, 835–839 (2016). URL <http://dx.doi.org/10.1038/nmat4623>.
- [12] Meevasana, W. et al. Creation and control of a two-dimensional electron liquid at the bare SrTiO₃ surface. Nat Mater **10**, 114–118 (2011). URL <http://dx.doi.org/10.1038/nmat2943>.
- [13] Santander-Syro, A. F. et al. Two-dimensional electron gas with universal subbands at the surface of SrTiO₃. Nature **469**, 189–193 (2011). URL <http://dx.doi.org/10.1038/nature09720>.
- [14] Lechermann, F. Unconventional electron states in δ -doped SmTiO₃. Scientific Reports **7**, 1565 (2017). URL <https://doi.org/10.1038/s41598-017-01847-5>.
- [15] Yoshimatsu, K. et al. Metallic quantum well states in artificial structures of strongly correlated oxide. Science **333**, 319 (2011). URL <http://science.sciencemag.org/content/333/6040/319.abstract>.
- [16] Kobayashi, M. et al. Origin of the anomalous mass renormalization in metallic quantum well states of strongly correlated oxide srvo₃. Phys. Rev. Lett. **115**, 076801 (2015). URL <https://link.aps.org/doi/10.1103/PhysRevLett.115.076801>.
- [17] King, P. D. C. et al. Quasiparticle dynamics and spin-orbital texture of the SrTiO₃ two-dimensional electron gas. Nature Communications **5**, 3414 (2014). URL <http://dx.doi.org/10.1038/ncomms4414>.
- [18] Chang, Y. J. et al. Layer-by-layer evolution of a two-dimensional electron gas near an oxide interface. Phys. Rev. Lett. **111**, 126401 (2013). URL <https://link.aps.org/doi/10.1103/PhysRevLett.111.126401>.
- [19] Plumb, N. C. et al. Mixed dimensionality of confined conducting electrons in the surface region SrTiO₃. Phys. Rev. Lett. **113**, 086801 (2014). URL <https://link.aps.org/doi/10.1103/PhysRevLett.113.086801>.
- [20] Moser, S. et al. How to extract the surface potential profile from the ARPES signature of a 2DEG. Journal of Electron Spectroscopy and Related Phenomena **225**, 16–22 (2018). URL <http://www.sciencedirect.com/science/article/pii/S0368204817301536>.
- [21] Chang, Y. J., Bostwick, A., Kim, Y. S., Horn, K. & Rotenberg, E. Structure and correlation effects in semiconducting srtio₃. Phys. Rev. B **81**, 235109 (2010). URL <https://link.aps.org/doi/10.1103/PhysRevB.81.235109>.
- [22] Zhou, H. D. & Goodenough, J. B. Localized or itinerant tio 3 electrons in rtio 3 perovskites. Journal of Physics: Condensed Matter **17**, 7395 (2005). URL <http://stacks.iop.org/0953-8984/17/i=46/a=023>.

Reviewers' comments:

Reviewer #1 (Remarks to the Author):

The authors carried out additional experiments using 6SrO sample and confirmed that the Lifshitz transition indeed takes places between 5SrO and 6SrO samples. They answered to all my questions. Therefore, I would like to recommend the publication of this work on Nature Communications.

However, I am still curious why the Lifshitz transition does not cause a discontinuity in the residual resistance, which is why the previous work, new Ref. 26, excluded such possibility. I would be interested in the authors' opinion about this discrepancy either in the paper or communication.

Reviewer #2 (Remarks to the Author):

Previously, I pointed out, as the major concern, that the results for 6-SrO (or 7-SrO) are missing in the manuscript, thus one cannot tell whether or not the jump of transport values between 5-SrO and 6-SrO really has a direct relation with the Lifshitz Transition of FS topology as claimed by the authors. In the new manuscript, the authors have added the data for 6-SrO, demonstrating that it has an electron-type FS centered at Gamma, and hence Lifshitz Transition indeed occurs between the border. The presentation for the energy location of VHS has also become improved by plotting the spectra, not only the ARPES image. The photon energy dependence shown in the supplementary figure also supports that the ARPES data are actually excited from the interface state. Because of these, I now agree that the transport anomaly coincides with the Lifshitz Transition of FS topology.

It was still not clear to me what is, in fact, the relation between the mass enhancement, Lifshitz Transition (location of VHS), and interlayer coupling. In the current version, the authors only mention that these are related, but do not detail how these are related. Moreover, since the SmTiO₃ is distorted, the SrTiO₃ layers adjacent to it should be more distorted for thinner samples. I am also concern about the thickness dependence of SmTiO₃ capping as discussed in Chang et al., PRL 111, 126401 (2013). The authors should address these issues to make the paper more solid and interesting. If the revised version could satisfy it, I will recommend the results for publication in Nature Communications.

Reply to the Report to Referee #1

The authors carried out additional experiments using 6SrO sample and confirmed that the Lifshitz transition indeed takes places between 5SrO and 6SrO samples. They answered to all my questions. Therefore, I would like to recommend the publication of this work on Nature Communications.

We thank the reviewer for his/her recommendation of our manuscript to be published on *Nature Communications*.

However, I am still curious why the Lifshitz transition does not cause a discontinuity in the residual resistance, which is why the previous work, new Ref. 26, excluded such possibility. I would be interested in the authors' opinion about this discrepancy either in the paper or communication.

Thank you for asking. The only discontinuity we observe is in the residual of the Hall angle. Although this should in principle reflect a change of the topology of the electronic structure, at this point it is not clear why we did not see a discontinuity also in the residual resistance. We hope that this work will stimulate further theoretical and experimental work to shed light on this important point.

Reply to the Report to Referee #2

Previously, I pointed out, as the major concern, that the results for 6SrO (or 7SrO) are missing in the manuscript, thus one cannot tell whether or not the jump of transport values between 5SrO and 6SrO really has a direct relation with the Lifshitz Transition of FS topology as claimed by the authors. In the new manuscript, the authors have added the data for 6SrO, demonstrating that it has an electron-type FS centered at Gamma, and hence Lifshitz Transition indeed occurs between the border. The presentation for the energy location of VHS has also become improved by plotting the spectra, not only the ARPES image. The photon energy dependence shown in the supplementary figure also supports that the ARPES data are actually excited from the interface state. Because of these, I now agree that the transport anomaly coincides with the Lifshitz Transition of FS topology.

We thank the reviewer for acknowledging the improvement of our manuscripts.

It was still not clear to me what is, in fact, the relation between the mass enhancement, Lifshitz Transition (location of VHS), and interlayer coupling. In the current version, the authors only mention that these are related, but do not detail how these are related. Moreover, since the SmTiO₃ is distorted, the SrTiO₃ layers adjacent to it should be more distorted for thinner samples.

When the distance of the interfaces is smaller than $t_{QW} \sim 6\text{-SrO}$, the two interfaces/wavefunctions start to couple, forming one quantum well. By decreasing the thickness, more electrons are confined in the narrower and flatter quantum potential, resulting in a significant increase of electrostatic doping level at the interface and each TiO₂ layer in SrTiO₃. This significantly higher electron doping in each TiO₂ layer leads to an enhancement of electron-electron correlation as evidenced by the strong mass renormalization in our measurements. Such enhancement of the correlation becomes a trigger of the Lifshitz transition as reported[1–3]. Our results suggest the competition/combination between the significant electrostatic doping by confinement and the renormalization from electron correlation makes the energy level of VHS fine tunable near E_F . In order to figure out the direct relation between Lifshitz transition and electronic correlation, further theoretical DFT+DMFT studies with different correlation effects (U) on different atomic sites are needed. However, such calculations are beyond our capabilities, and beyond the main point and conclusion in this work. We hope that this work will stimulate such type of theoretical work. Regarding the distortion, we observed Fermi surface folding ($\sqrt{2} \times \sqrt{2} \times 45^\circ$) following the same manner for all systems in this study, suggesting that the effect from the distortion at the interface for $t_{QW} \geq 3\text{-SrO}$ are mostly the same. As shown in the previous work[4], the 2-SrO quantum well in SmTiO₃, which can be considered to have more distortion than any of the samples in our work, shows only a very slight increase in the deviation angle and a small difference of deviation angle from 5-SrO. Therefore, we believe we could effectively ignore the difference of the distortion effects between the samples ($t_{QW} \geq 3\text{-SrO}$) in our work. We revised the main text accordingly.

I am also concern about the thickness dependence of SmTiO₃ capping as discussed in Chang et al., PRL 111, 126401 (2013).

We would like to clarify that the work by Chang *et al.* is completely different from ours and this concern is not relevant to our work. Chang *et al.* investigated the thickness dependent of SrTiO₃ capped on LaTiO₃[5] and they observed the electronic structure evolution depending on the thickness of capping SrTiO₃ layer. In their measurement, the charge source, LaTiO₃, is embedded in the carrier acceptor (SrTiO₃). By changing the thickness of the charge acceptor SrTiO₃, they could investigate how the doped charges are distributed in SrTiO₃, resulting in the “evolution” of the electronic structure layer by layer[6, 7]. On the other hand, in our work, the capping layer SmTiO₃ plays a role to be a charge source and also a potential barrier to create the confined electronic states in SrTiO₃, and hence conducting electrons are not distributed in SmTiO₃[8]. If we measure the electronic structures of different thickness of SmTiO₃ capping layer with ARPES, we simply lose the intensity from the interface and start to see the electronic structure of the bulk SmTiO₃ instead. Unlike Chang *et al.* PRL where the charge distribution in the quantum well was investigated, our work focuses only on the interface. We have now added the description about this aspect in the main manuscript.

The authors should address these issues to make the paper more solid and interesting. If the revised version could satisfy it, I will recommend the results for publication in Nature Communications.

We thank again the referee for the comments and suggestions.

-
- [1] Liu, D.-Y., Sun, Z., Lu, F., Wang, W.-H. & Zou, L.-J. Correlation-driven lifshitz transition in electron-doped iron selenides (li,fe)ohfese. Phys. Rev. B **98**, 195137 (2018). URL <https://link.aps.org/doi/10.1103/PhysRevB.98.195137>.
 - [2] Xu, N. et al. Evidence of a coulomb-interaction-induced lifshitz transition and robust hybrid weyl semimetal in T_d -mote₂. Phys. Rev. Lett. **121**, 136401 (2018). URL <https://link.aps.org/doi/10.1103/PhysRevLett.121.136401>.
 - [3] Skorniyakov, S. L. & Leonov, I. Correlated electronic structure, orbital-dependent correlations, and lifshitz transition in tetragonal fes (2019). URL <https://ui.adsabs.harvard.edu/abs/2019arXiv190512244S>.
 - [4] Zhang, J. Y. et al. Correlation between metal-insulator transitions and structural distortions in high-electron-density SrTiO₃ quantum wells. Phys. Rev. B **89**, 075140 (2014). URL <https://link.aps.org/doi/10.1103/PhysRevB.89.075140>.
 - [5] Chang, Y. J. et al. Layer-by-layer evolution of a two-dimensional electron gas near an oxide interface. Phys. Rev. Lett. **111**, 126401 (2013). URL <https://link.aps.org/doi/10.1103/PhysRevLett.111.126401>.
 - [6] Larson, P., Popovi, Z. S. & Satpathy, S. Lattice relaxation effects on the interface electron states in the perovskite oxide heterostructures: latio₃ monolayer embedded in srtio₃. Phys. Rev. B **77**, 245122 (2008). URL <https://link.aps.org/doi/10.1103/PhysRevB.77.245122>.
 - [7] Popovic, Z. S. & Satpathy, S. Wedge-shaped potential and airy-function electron localization in oxide superlattices. Phys. Rev. Lett. **94**, 176805 (2005). URL <https://link.aps.org/doi/10.1103/PhysRevLett.94.176805>.

- [8] Bjaalie, L. et al. Band alignments between SmTiO₃, GdTiO₃, and SrTiO₃. Journal of Vacuum Science & Technology A: Vacuum, Surfaces, and Films **34**, 061102 (2016).
URL <http://dx.doi.org/10.1116/1.4963833>.